# Vaccination for Monkeypox Virus Infection in Humans: A Review of Key Considerations

**DOI:** 10.3390/vaccines10081342

**Published:** 2022-08-18

**Authors:** Kay Choong See

**Affiliations:** Division of Respiratory & Critical Care Medicine, Department of Medicine, National University Hospital, Singapore 119228, Singapore; kaychoongsee@nus.edu.sg

**Keywords:** epidemiology, infection control, post-exposure prophylaxis, pre-exposure prophylaxis, smallpox, vaccinia virus

## Abstract

Monkeypox virus infection in humans (MVIH) is currently an evolving public health concern given that >3000 MVIH cases have been reported in >50 countries globally, and the World Health Organization declared monkeypox a global health emergency on 23 July 2022. Adults (≥16 years old) usually have mild disease in contemporary studies, with a pooled case fatality rate of 0.03% (1/2941 cases). In comparison, poorer outcomes have been reported in children <16 years old (pooled case fatality rate 19% (4/21 cases)), immunocompromised patients, and pregnant women, with high rates of fetal demise in this group. Monkeypox-specific treatments include oral or intravenous tecovirimat, intravenous or topical cidofovir, oral brincidofovir, and vaccinia immunoglobulin, but the overall risk–benefit balance of monkeypox-specific treatment is unclear. Two effective vaccines exist for the prevention of MVIH: modified vaccinia Ankara and ACAM2000. Most probably, vaccination will be a key strategy for mitigating MVIH given the current rapid global spread of monkeypox, the existence of efficacious vaccines, and the uncertain risk–benefit profile of current antivirals. Priority groups for vaccination should include healthcare workers at high risk for occupational exposure, immunocompromised patients, and children. Vaccination strategies include pre-exposure vaccination, post-exposure prophylaxis, and ring vaccination of close contacts.

## 1. Introduction

Monkeypox is a zoonotic orthopox DNA virus that is in the same genus as the variola virus (which causes smallpox) and the vaccinia virus (which is used in smallpox vaccines). Following the first reported case in a 9-month-old boy [1], monkeypox had previously caused only sporadic outbreaks of human disease within Africa, with limited secondary spread of travel-associated cases outside Africa [2,3]. However, monkeypox virus infection in humans (MVIH) is currently an evolving public health concern given that >3000 MVIH cases have been reported in >50 countries globally [4]. Increased reporting of MVIH is expected due to the cessation of smallpox vaccination since the eradication of smallpox in 1980 (smallpox vaccination has cross-protection for MVIH [5]), waning immunity in previously vaccinated adults [6], and cross-border international travel. Owing to the significant and rapid global spread of monkeypox, the World Health Organization declared monkeypox a global health emergency on 23 July 2022.

Containment of the current monkeypox global health emergency will not be easy. Four decades after the cessation of smallpox vaccination, global orthopox virus immunity is low, allowing monkeypox to spread relatively unchecked. Eradication of monkeypox may also be challenging since monkeypox, unlike smallpox, has non-human reservoirs of disease. Meanwhile, the virulence of monkeypox should not be underestimated. Next to smallpox, monkeypox is the second most pathogenic orthopox virus in humans [1]. Genetic variation among monkeypox strains may lead to immune-modulating genes that inhibit host defenses. For instance, the monkeypox inhibitor of complement enzymes (MOPICE) gene is present in Clade 1 but not in Clade 2, contributing to more severe disease and mortality in patients infected by Clade 1 [7]. Whatever the clade, smallpox-naïve and unvaccinated individuals will be disproportionately affected by MVIH. For instance, among 282 patients evaluated in Zaire between 1980 and 1985, the case fatality rate was higher in unvaccinated individuals compared to individuals vaccinated against smallpox (11% versus 0%) [8].

In general, vaccination is an important strategy for mitigating both the spread and severity of transmissible viral infections, particularly for immunocompromised individuals [9]. For MVIH, whether vaccination is warranted requires an understanding of the severity of the infection (pathology), risk of acquiring infection (epidemiology), treatment options, and prevention options. This paper therefore aims to review the contemporary literature about the pathology, epidemiology, treatment, and prevention of human monkeypox infection. A narrative synthesis of these key considerations will be undertaken.

## 2. Materials and Methods

The PubMed database (pubmed.ncbi.nlm.nih.gov) was searched for contemporary articles from 1 January 2017 to 22 July 2022 using the keyword “monkeypox”, resulting in 391 articles, with 238 (60.9%) articles published in 2022 alone. Original clinical and outcome data were extracted from published reports and organized according to World Health Organization administrative regions (Table 1).

## 3. Results

### 3.1. Pathology

MVIH is defined by a positive result on a monkeypox virus polymerase chain reaction (PCR) assay in a specimen from any anatomical site [4,10]. Clinical samples that have demonstrated positivity include skin/anogenital lesions, blood, saliva, nose/throat/nasopharyngeal swab, semen, urine, rectal swab, feces, and deep tissue abscesses [4,11,12]. At least two clades of monkeypox exist: Clade 1 (associated with the Congo Basin) and Clade 2 (associated with West Africa). A new clade, Clade 3, has been suggested after phylogenomic analysis of the current 2022 European outbreak [13].

After a median incubation period of 7 days (range 3–20 days), MVIH presents clinically with a vesiculopustular illness, fever, and lymphadenopathy that may persist for another 3–4 weeks [4,10]. Skin lesions can be particularly extensive in immunocompromised patients, such as persons living with the human immunodeficiency virus [14]. Complications of MVIH include encephalitis, myocarditis, pneumonitis, acute kidney injury, keratitis, and secondary bacterial infections [4,11], which are again seen more in immunocompromised patients [15,16].

Adults (≥16 years old) usually have mild disease in contemporary studies [4,11], with a pooled case fatality rate of 0.03% (1/2941 cases) (Table 2), excluding the three studies with mixed adult–pediatric populations [15,16,17]. In comparison, serious consequences have been reported in children <16 years old [18] (case fatality rate of 17% in an older study [19] and 19% (4/21 cases) from the pooled results in Table 3), and in immunocompromised patients (case fatality rates up to 11% [20]). Prolonged PCR positivity of the skin lesions beyond 2 weeks has been reported in children, which may indicate a persistent presence of a replication-competent virus and increased propensity for transmission (Table 3). Separately, pregnancy outcomes appear to be poor, with high rates of fetal demise [6] (Table 4).

### 3.2. Epidemiology

Although monkeypox was first found in monkeys, monkeys are incidental hosts, and the animal reservoir remains unknown. Animal-to-human transmission, human-to-human transmission, and asymptomatic circulation of monkeypox in human communities can occur [49,50]. Spread of monkeypox virus from forest animals (e.g., monkeys, rodents, squirrels) to exposed humans has been reported [19], which may be via contact with bodily fluids, bites, or preparation of bushmeat. Spread of monkeypox virus among humans occurs via close contact (including sexual contact), fomites, environmental contamination [51], respiratory droplets, or vertical (maternal–fetal) transmission [4,6].

Both travel-related [27,28,44] and autochthonous community transmission [36,52] have been reported. Secondary attack rates in contacts unvaccinated against smallpox are estimated to be about 10% [20]. Prior childhood vaccination is not completely protective, as 9% of cases reported prior vaccination in the largest observational study of the 2022 outbreak to date [4].

### 3.3. Treatment Options

Monkeypox-specific treatments include oral or intravenous tecovirimat (a small molecule virus inhibitor), intravenous or topical cidofovir (a viral DNA polymerase inhibitor), oral brincidofovir (a prodrug of cidofovir), and vaccinia immunoglobulin [4,11,53] (Table 5). Tecovirimat can be used for adults and children weighing at least 3 kg [54]. For pregnant women with severe disease, tecovirimat and vaccinia immunoglobulin are preferred over cidofovir/brincidofovir as cidofovir is a teratogen. 

In one case report, one adult patient received tecovirimat and recovered after one month in hospital [24]. In a case series, three adult patients treated with oral brincidofovir (200 mg once a week) all developed elevated liver enzymes, resulting in therapy cessation, and one patient treated with oral tecovirimat (200 mg twice daily for 2 weeks) experienced no adverse effects [11]. Nonetheless, the overall risk–benefit balance of monkeypox-specific treatment is unclear.

### 3.4. Prevention Options

Smallpox (vaccinia) vaccination is cross-protective for both smallpox and monkeypox [55], preventing about 85% of MVIH [56], though vaccine supply and production capacity are currently limited. Two vaccines exist: modified vaccinia Ankara (Jynneos/Imamune/Imvanex, Bavarian Nordic, Hørsholm, Denmark) and ACAM2000 (Emergent BioSolutions, Gaithersburg, MD, USA) (Table 6). The modified vaccinia Ankara vaccine is made from a highly attenuated and nonreplicating vaccinia virus and can be used in immunocompromised patients. It is administered as two subcutaneous injections separated by 4 weeks, with the peak antibody response 2 weeks after completion of the two-dose series. On the other hand, ACAM2000 contains replication-competent smallpox (vaccinia) and is contraindicated in immunocompromised and pregnant patients. It is administered once by a multiple-puncture technique using a bifurcated needle, with the peak antibody response after 28 days. After vaccination, local side effects (pain, erythema, induration) and systemic side effects (fever, chills, fatigue, myalgia, headache, nausea) are generally mild and transient. Nevertheless, severe adverse events (post-vaccination encephalitis and myopericarditis) may rarely occur, particularly for ACAM2000 [55]. Regular booster doses are recommended for persons with ongoing occupational risk for exposure to monkeypox (every 2 years for Jynneos and every 3 years for ACAM2000) [57].

Given the limited vaccine supplies, prioritization of the following groups for pre-exposure vaccination may be reasonable: healthcare workers at high risk for occupational exposure [29,57], immunocompromised patients, and children (given the increased risk of severe disease and death). Despite the increased risk of pregnancy loss and stillbirth with MVIH, live vaccines should be used with caution in pregnancy, with the replication-incompetent modified vaccinia Ankara being preferred [54]. While vaccination is best administered before infection, vaccination as a post-exposure prophylaxis may also be useful if given within 4 days of exposure. This is especially important for high-risk exposures, such as physical contact with an infected person’s broken skin/mucous membrane or being near an infected person during an aerosol-generating procedure while not wearing a surgical mask/respirator [24]. Additionally, for epidemic containment, besides rapid diagnostic testing [58] and prompt identification and isolation of cases, vaccination of close contacts (ring vaccination) can help [59].

To limit the spread of monkeypox to both vaccinated and non-vaccinated persons, non-pharmaceutical shielding methods should address the known transmission routes. These methods include isolation of infected persons, hand hygiene, wearing of appropriate personal protection (long-sleeved gowns, gloves), proper cleaning of contaminated surfaces (for instance, with 70% ethanol [60]), and disposal of fomites. These methods can limit direct and indirect contact transmission. In addition, the wearing of face masks by infected persons and their close contacts can limit droplet transmission. When aerosol-generating procedures are performed for MVIH cases, healthcare staff may consider a higher level of protection, with N95/filtering facepiece 3 (FFP3) respirator masks for respiratory protection and face shields/goggles for eye protection [29].

## 4. Discussion

Contemporary data on MVIH suggest that the disease is mild in adults and more severe in children. Outcomes for immunocompromised patients and pregnant women may also be poor, with a high risk of fetal loss in the latter. Transmission risk is substantial from close contact and droplet spread, exacerbated by international travel. While specific treatment is limited, two current vaccinia virus-based vaccines are known to prevent MVIH effectively.

Although MVIH is not a novel disease, several knowledge gaps exist. Firstly, the current global outbreak is larger and more extensive than before, and the full impact of disease on the wider population of older persons, children, immunocompromised patients, pregnant mothers, and patients from resource-limited settings remains uncertain. Secondly, virulence factors contributing to the variable severity and outcomes of monkeypox infection have not been fully elucidated [7]. Thirdly, monitoring for post-MVIH sequelae would be needed, especially for serious complications such as sight-threatening corneal opacities and physically debilitating scars [8,61].

In addition, transmission routes are not completely clear, and more work is required to uncover sources and methods of transmission, including the extent to which airborne transmission contributes to the current monkeypox outbreak. Given that monkeypox can infect animals and animal reservoirs are not fully known [58,61], disease prevalence among animals in general—including domestic pets—should be studied. On a related note, measures to mitigate viral spread among animals should be considered for zoonotic transmission.

As case numbers increase, public health resources will be stretched. Increasing patient numbers may also reveal more cases with serious manifestations and complications, further burdening healthcare systems. Additional studies are needed to elucidate the efficacy and safety of current antiviral treatments and vaccines. More research is also needed to develop novel diagnostics (e.g., lateral flow assay for the detection of orthopoxviruses [62]) and therapeutics. In the meantime, frontline healthcare staff around the world should be trained to rapidly identify, isolate, and manage patients with MVIH [5].

## 5. Conclusions

Most probably, vaccination will be a key strategy for mitigating MVIH given the current rapid global spread of monkeypox, the existence of efficacious vaccines, and the uncertain risk–benefit profile of current antivirals. Priority groups for vaccination should include healthcare workers at high risk for occupational exposure, immunocompromised patients, and children. In addition, vaccination as post-exposure prophylaxis, ring vaccination of close contacts, and non-pharmaceutical shielding methods may also be used to limit human-to-human spread [10].

## Figures and Tables

**Table 1 vaccines-10-01342-t001:** List of WHO regions.

Region	Countries
African Region (AFR)	Algeria, Angola, Benin, Botswana, Burkina Faso, Burundi, Cameroon, Cape Verde, Central African Republic, Chad, Comoros, Ivory Coast, Democratic Republic of the Congo, Equatorial Guinea, Eritrea, Ethiopia, Gabon, Gambia, Ghana, Guinea, Guinea-Bissau, Kenya, Lesotho, Liberia, Madagascar, Malawi, Mali, Mauritania, Mauritius, Mozambique, Namibia, Niger, Nigeria, Republic of the Congo, Rwanda, São Tomé and Príncipe, Senegal, Seychelles, Sierra Leone, South Africa, South Sudan, Eswatini, Togo, Uganda, Tanzania, Zambia, Zimbabwe
Region of the Americas (AMR)	Antigua and Barbuda, Argentina, Bahamas, Barbados, Belize, Bolivia, Brazil, Canada, Chile, Colombia, Costa Rica, Cuba, Dominica, Dominican Republic, Ecuador, El Salvador, Grenada, Guatemala, Guyana, Haiti, Honduras, Jamaica, Mexico, Nicaragua, Panama, Paraguay, Peru, Saint Kitts and Nevis, Saint Lucia, Saint Vincent and the Grenadines, Suriname, Trinidad and Tobago, United States, Uruguay, Venezuela
Eastern Mediterranean Region (EMR)	Afghanistan, Bahrain, Djibouti, Egypt, Iran, Iraq, Jordan, Kuwait, Lebanon, Libya, Morocco, Oman, Pakistan, Palestine, Qatar, Saudi Arabia, Somalia, Sudan, Syria, Tunisia, United Arab Emirates, Yemen
European Region (EUR)	Albania, Andorra, Armenia, Austria, Azerbaijan, Belarus, Belgium, Bosnia and Herzegovina, Bulgaria, Croatia, Cyprus, Czech Republic, Denmark, Estonia, Finland, France, Georgia, Germany, Greece, Hungary, Iceland, Ireland, Israel, Italy, Kazakhstan, Kyrgyzstan, Latvia, Lithuania, Luxembourg, Malta, Moldova, Monaco, Montenegro, Netherlands, North Macedonia, Norway, Poland, Portugal, Romania, Russia, San Marino, Serbia, Slovakia, Slovenia, Spain, Sweden, Switzerland, Tajikistan, Turkey, Turkmenistan, Ukraine, United Kingdom, Uzbekistan
South-East Asian Region (SEAR)	Bangladesh, Bhutan, North Korea, India, Indonesia, Maldives, Myanmar, Nepal, Sri Lanka, Thailand, Timor-Leste
Western Pacific Region (WPR)	Australia, Brunei, Cambodia, China, Cook Islands, Fiji, Japan, Kiribati, Laos, Malaysia, Marshall Islands, Micronesia, Mongolia, Nauru, New Zealand, Niue, Palau, Papua New Guinea, Philippines, Samoa, Singapore, Solomon Islands, South Korea, Tonga, Tuvalu, Vanuatu, Vietnam

**Table 2 vaccines-10-01342-t002:** Adult cases of monkeypox virus infection in humans.

WHO Region	N	Patient Characteristics	Source of Infection	Clinical Features	Treatment and Outcomes	Ref/Year
AFR	7	24–41 years old, 2 (29%) males	Transport industry and healthcare contact	Vesiculo-papular rash (100%), fever (86%), cervical and inguinal lymphadenopathy (86%)	Supportive treatment, recovery without sequelae	[21] 2017
AFR	26	Median age 24 years, 14 (53.8%) males ^a^	Hunting, bushmeat, family contact	Fever (100%), rash (100%), pruritus (46%), cervical and inguinal lymphadenopathy (35%)	61% hospitalized, 2 (7.7%) deaths	[17] 2018
AFR	8	24–40 years old, 3 (38%) males	Hunting, bushmeat, family contact	Fever, rash	Supportive treatment, 1 death	[22] 2019
AFR	122	Median age 29 years, 84 (69%) males ^a^	Zoonotic, human-to-human, 2 healthcare-associated	Vesiculopustular rash (100%), fever (88%), headache (79%), pruritus (73%), lymphadenopathy (69%), myalgia (63%), sore throat (58%)	1 spontaneous 2nd-trimester abortion in pregnant woman, supportive care for monkeypox, 7/118 (6%) deaths ^b^	[16] 2019
AFR	40	Median age 32 years, 31 (78%) males, 9 with HIV ^a^	Unclear	Rash (100%), fever (90%), lymphadenopathy (88%), genital ulcer (63%), body ache (63%), headache (48%), sore throat (45%), pruritus (38%), conjunctivitis and photophobia (23%), nasal congestion (13%), cough (13%), skin ulcers (13%), nausea/vomiting (8%), hepatomegaly (8%), scrotal edema (5%)	21 (53%) developed complications (19 bacterial skin infection, 5 gastro-enteritis, 4 sepsis, 3 pneumonia, 3 encephalitis, 3 keratitis, 1 premature rupture of membrane at 2nd trimester with intrauterine fetal death), 11 (28%) developed anxiety and depression requiring counselling, supportive care for monkeypox, 5 (13%) deaths ^c^	[15] 2020
AFR	2	20-year-old brothers	Family contact	Fever, headache, odynophagia, dysuria, generalized skin lesions	Supportive treatment, recovered	[23] 2021
AMR, EUR, WPR	528	Median age 38 years old, 75% White, 98% gay/bisexual men	Sexual close contact (95%)	Rash (95%), anogenital lesions (73%), fever (62%), lymphadenopathy (56%), mucosal lesions (41%), lethargy (41%), myalgia (31%), headache (27%)	Hospitalized for severe pain, soft-tissue superinfection, severe pharyngitis, eye lesions, acute kidney injury, myocarditis, infection control (13%), 5% received monkeypox-specific treatment (12 cidofovir, 8 tecovirimat, 1 vaccinia IVIG), no deaths	[4] 2022
AMR	17	28–61 years old	Unclear	Rash (100%), fatigue or malaise (76%), chills (71%), lymphadenopathy (53%), headache (47%), fever (41%), body ache (35%), sore throat or cough (29%), sweat (24%)	1 patient treated with tecovirimat, 100% recovered	[10] 2022
AMR	1	Middle-aged man	Unclear	Rash, fever, cough, fatigue, diarrhea, vomiting	Treated with tecovirimat, recovered	[24] 2022
AMR	1	33-year-old man	Sexual contact	Rectal pain, generalized umbilicated and pustular rash, tender cervical and inguinal lymphadenopathy	Supportive treatment, recovered	[25] 2022
AMR	1	28-year-old man	Unclear	Diffuse vesicular rash, umbilicated pustules, cervical lymphadenopathy	Supportive treatment, self-resolving	[26] 2022
EUR	2	Adult men	Unclear	Fever, rash, lymphadenopathy	Supportive treatment, self-resolving	[27] 2018
EUR	1	38-year-old man	Unclear	Fever, chills, rash, itchy penile ulcers, tender inguinal lymphadenopathy	Supportive treatment, self-resolving	[28] 2019
EUR	6	30–50 years old, 4 men, 2 women	Household contact; nosocomial	Rash (100%), lymphadenopathy (67%), reactive low mood (50%), deep thigh abscess (17%)	3 patients treated with brincidofovir, 1 with tecovirimat, prolonged PCR positivity 26–39 days, no deaths	[11] 2022
EUR	2	1 adult male patient, 1 healthcare worker taking care of the patient	1 travel-related, 1 work-related contact	Rash, fever, lymphadenopathy, malaise, headache, sore throat, earache, eye pain	Supportive treatment, recovered	[29] 2020
EUR	2	1 adult man and 1 adult woman	Family contact	Vesicular lesion	Supportive treatment, recovered	[2] 2021
EUR	12	32–52 years old, all men	2 contact, 10 unclear	Rash (100%), non-specific syndrome of fever/myalgia/ malaise (92%)	Supportive treatment, no deaths	[12] 2022
EUR	1	71-year-old woman	Sexual contact	Extensive maculopapular rash, fever, fatigue, drowsiness, umbilicated skin lesions, inguinal lymphadenopathy	Supportive treatment, discharged in good condition after 3 days of hospitalization	[30] 2022
EUR	48	Median 35 years old (IQR 29–44), all men	Sexual contact	Vesicular-umbilicated skin lesions (94%), asthenia (67%), painful inguinal lymphadenopathy (52%), fever (52%), myalgia (52%), headache (52%), proctitis (27%), cough (17%), urethritis (15%)	Supportive treatment, 1 patient required hospitalization for 32h, all recovered without sequelae	[31] 2022
EUR	1	24-year-old man with HIV	Sexual contact	Painful skin umbilicated papules and pustules, painful tongue ulcer, inguinal and cervical lymphadenopathy, fatigue, anal pain, fever	Supportive treatment, discharged after 5 days in hospital	[14] 2022
EUR	1	44-year-old man with HIV	Sexual contact	Painless multiple vesiculopustular skin lesions with umbilication and central crusting	Supportive treatment with recovery after 1 week	[32] 2022
EUR	1255	Median age 37 years old, 98.9% male	Intimate and prolonged contact during sex (86%)	Anogenital rash (67%), fever (57%), disseminated rash (55%), lymphadenopathy (50%), asthenia (42%), oral/buccal rash (17%), myalgia (32%), throat pain (26%), headache (26%)	30 (6%) of the 530 cases were hospitalized (median admission 2 days), 33 reported complications (15 secondary bacterial infections, 11 oral ulcers, 2 proctitis, 2 pharyngotonsillitis), no deaths	[33] 2022
EUR	2	26 and 32 years old, both men	Sexual contact	Skin papules and pustules, malaise, body ache, inguinal lymphadenopathy, dysphagia, fatigue, cough	Supportive treatment, recovered	[34] 2022
EUR	508	18–67 years old, 503 (99%) men, 225 (44%) had HIV	Sexual contact (84%)	Exanthema (98%), anogenital and perineal rash (72%), lymphadenopathy (61%), asthenia (47%), myalgia (36%), headache (32%), odynophagia (28%), proctitis (16%)	Supportive treatment, 19 (3.7%) required hospitalization, with 7 having parapharyngeal abscesses, mouth ulcers, and bacterial superinfection, no deaths	[35] 2022
EUR	521	20–67 years old, all men	Sexual or intimate contact	Symptoms not specified	38 (8%) of 455 cases hospitalized, no deaths	[36] 2022
EUR	1	42-year-old man	Sexual contact	Cutaneous and perianal lesions, rectal ulcers	Concurrent infection with *Campylobacter* spp., supportive treatment, recovered	[37] 2022
EUR	1	32-year-old man with HIV	Sexual contact	Umbilicated pustules over genitals and hands, painful inguinal lymphadenopathy	Supportive treatment, recovered	[38] 2022
EUR	1	Adult man	Sexual contact	Fever, intense fatigue, chills, myalgia, sore throat, severe anal pain, no rash	Supportive treatment, recovered	[39] 2022
EUR	1	26-year-old man with HIV	Sexual contact	Fever, chills, rectal pain, umbilicated vesiculopustular rash, dysphagia, lymphadenopathy	Supportive treatment, recovered	[40] 2022
EUR	2	Both 37-year-old men	Sexual contact	Genital lesions (papules, vesicles), fever	Supportive treatment, recovered	[41] 2022
EUR	1	33-year-old man with HIV	Sexual contact	Asthenia, malaise, anorexia, fever, papules on elbows, ulcerated perianal lesion, inguinal lymphadenopathy	Supportive treatment, recovered	[42] 2022
EUR	4	All males in their 30s, 2 with HIV	Sexual contact	Umbilicated skin lesions in genital areas and limbs, inguinal lymphadenopathy, fever	Supportive treatment, recovered	[43] 2022
WPR	1	39-year-old man	Unclear	Fever, chills, myalgia, vesicular and pustular rash, cervical and inguinal lymphadenopathy	Supportive treatment, recovered	[44] 2020
WPR	1	34-year-old man	Unclear	Painless genital lesion, tender inguinal lymphadenopathy, fever, sore throat, skin rash	Supportive treatment, no death	[45] 2022
WPR	1	Man in his 30s with HIV	Sexual contact	Painless white pustules on the penis that became painful and pruritic, truncal rash, fever, malaise	Supportive treatment, recovering after 2 days in hospital	[46] 2022

^a^ Combination of adults and children. ^b^ One neonatal death, four patients who died also had HIV with acquired immunodeficiency syndrome, and two other deaths were from secondary bacterial infection of skin lesions. ^c^ One neonatal pneumonia with encephalitis, one adult encephalitis, two adult sepsis, one adult suicide. HIV: human immunodeficiency virus. IQR: interquartile range. IVIG: intravenous immunoglobulin. PCR: polymerase chain reaction. Ref: reference. World Health Organization regions: African Region (AFR), Region of the Americas (AMR), South-East Asian Region (SEAR), European Region (EUR), Eastern Mediterranean Region (EMR), Western Pacific Region (WPR).

**Table 3 vaccines-10-01342-t003:** Pediatric cases of monkeypox virus infection in humans.

WHO Region	N	Patient Characteristics	Source of Infection	Clinical Features	Outcomes	Ref/ Year
AFR	3	15 months–9 years old, all males	Family contact	Fever, vesiculo-papular rash, cervical adenitis	Supportive treatment, 2 (67%) deaths (signs of pulmonary edema, agitation, hypotonia)	[21] 2017
AFR	14	1–15 years old, 5 males	Family contact	Fever, rash	Supportive treatment, 2 (1.5%) deaths	[22] 2019
AFR	1	11-month-old boy	Unclear	Fever, chills, diaphoresis, vomiting, cough, pruritus, generalized umbilicated pustules	Supportive treatment, spontaneous recovery	[47] 2019
EUR	1	18-month-old child	Family contact	Vesicular lesion	Supportive treatment, recovered	[2] 2021
EUR	1	<2 years old, female	Unclear	Rash, lymphadenopathy	Supportive treatment, prolonged PCR positivity 22 days, no death	[11] 2022
EUR	1	<10 years old, male	Unclear	Rash, sore throat, incidental IgA deficiency	Supportive treatment, prolonged PCR positivity and partial resolution 3 weeks from symptom onset	[48] 2022

Ref: reference. World Health Organization regions: African Region (AFR), Region of the Americas (AMR), South-East Asian Region (SEAR), European Region (EUR), Eastern Mediterranean Region (EMR), Western Pacific Region (WPR).

**Table 4 vaccines-10-01342-t004:** Pregnancy-related monkeypox virus infection in humans.

WHO Region	N	Patient Characteristics	Source of Infection	Clinical Features	Outcomes	Ref/ Year
AFR	4	20–29 years old	Unclear	Vesiculopustular lesions	2 first-trimester miscarriages, 1 stillborn, 1 birth of healthy infant	[6] 2017

Ref: reference. World Health Organization regions: African Region (AFR), Region of the Americas (AMR), South-East Asian Region (SEAR), European Region (EUR), Eastern Mediterranean Region (EMR), Western Pacific Region (WPR).

**Table 5 vaccines-10-01342-t005:** Treatment and prevention of monkeypox virus infection.

Treatment	Prevention
Oral/intravenous tecovirimat, suitable for adults and children weighing >3 kg	Respiratory droplet protection with surgical masks, use respirators for aerosol-generating procedures
Intravenous/topical cidofovir or oral brincidofovir, avoid in pregnancy (teratogenic)	Hand hygiene, appropriate contact precautions including use of long-sleeved gowns and gloves during patient contact
Vaccinia immunoglobulin	Modified vaccinia Ankara or ACAM2000 vaccine

**Table 6 vaccines-10-01342-t006:** Monkeypox virus vaccines and their uses.

	Modified Vaccinia Ankara (Jynneos/Imamune/Imvanex, Bavarian Nordic, Hørsholm, Denmark)	ACAM2000 (Emergent BioSolutions, Gaithersburg, MD, USA)
Replication-competent	No	Yes
Administration	Subcutaneous	Multiple-puncture technique with a bifurcated needle
Number of doses	2, spaced 28 days apart	1
Peak vaccine response	2 weeks after 2nd dose	28 days after vaccination
Booster doses for ongoing high-risk (e.g., occupational) exposure	Every 2 years	Every 3 years
Pre-exposure vaccination	Yes	Yes
Post-exposure prophylaxis	Yes	Yes
Ring vaccination of close contacts for epidemic containment	Yes	Yes
Use in children	Yes	Yes
Use in pregnant women	Yes	No
Use in immunocompromised persons	Yes	No
Side effects	Usually mild, local, rarely severe (e.g., post-vaccination encephalitis and myopericarditis)	Usually mild, local, rarely severe (e.g., post-vaccination encephalitis and myopericarditis)

## Data Availability

The data presented in this study are contained within the article.

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
