# Peer review of "Vaccination for Monkeypox Virus Infection in Humans: A Review of Key Considerations"

_vaccines, 2022, doi:10.3390/vaccines10081342_

Round 1

Reviewer 1 Report

Dear Editor,

Thank you for the opportunity to review the manuscript Vaccination for monkeypox virus infection in humans: A re-2 view of key considerations This review identified vaccination as a key strategy for mitigating MVIH given the current rapid global spread of monkeypox because of  the existence of efficacious vaccines and the uncertain risk-benefit profile of current antivirals. Priority  groups for vaccination should include healthcare workers at high risk for occupational exposure, immunocompromised patients and children. Vaccination strategies include pre-exposure vaccination, post-exposure prophylaxis and ring vaccination of close contacts.  Although MVIH is not a novel disease, several knowledge gaps exist. The current global outbreak is larger and more extensive than before, and the full impact of disease  on the wider population of older persons, children, immunocompromised patients, pregnant mothers  remains uncertain. This review  is very interesting, well written,  informative and  concise there are no novel findings but provides a good literature review. It is my opinion that  can be considered for publication.

Author Response

Thank you for reviewing the manuscript and for your support.

Reviewer 2 Report

This review is devoted to the Monkeypox virus infection in humans that is now spreading throughout the world. Several cases are reported in nearly all countries in all parts of the world. That is why WHO has declared monkeypox a global health emergency on 23 July 2022. Author gives us a comprehensive and very interesting analysis of the monkey pox epidemic in several countries and in different groups of patients including various age intervals, and also in immunocompromized and pregnant. Monkeypox specific treatments are discussed as well as vaccination strategies.

Manuscript’s strengths lays in deep analysis of monkeypox epidemiology, in analysis of current treatment possibilities and their comparative efficacy in different gpoups of patients, and in the discussion on vaccination strategies using available vaccines.

I could not point out any weakness of this review. It is really comprehensive.

Author Response

Thank you for reviewing the manuscript and for your kind comments.

Reviewer 3 Report

The author presents a succinct, informative and useful review that gathers together data from a number of papers published in recent years on human monkeypox.

The review presents data in a quite readable form with clear Tables.

Some studies cited contain large datasets (eg 1255 patients – Rodriguez et al 2022) with regional symptoms and severity indicated by %. These larger datasets are useful in enabling readers to reliably assess the scope of the presentations, their severity, involvement of secondary infections and the treatment outcomes. Many of the studies cited contain reports of monkeypox infections in single patient reports. While together they provide useful information on clinical cases, treatment, treatment outcomes and geographical spread thus adding to the comprehensive coverage of the condition, they are arguable less informative than the few larger studies reported.

The introduction could be usefully extended by the inclusion of several other well cited papers on human monkeypox such as:

Brown, K., & Leggat. PA. "Human monkeypox: current state of knowledge and implications for the future." Tropical medicine and infectious disease 1.1 (2016): 8.

Damon, I. K. "Status of human monkeypox: clinical disease, epidemiology and research." Vaccine 29 (2011): D54-D59

Ježek, Z., Szczeniowski, M., Paluku, K. M., & Mutombo, M. (1987). Human monkeypox: clinical features of 282 patients. Journal of infectious diseases156(2), 293-298.

Ježek, Z., & Frank Fenner F., Human monkeypox. Monographs in Virology Vol. 17. S. Karger AG, 1988.

Sklenovska, N., & Van Ranst. M. "Emergence of monkeypox as the most important orthopoxvirus infection in humans." Frontiers in public health 6 (2018): 241.

Weaver, J. R., & Isaacs, S. N. (2008). Monkeypox virus and insights into its immunomodulatory proteins. Immunological reviews225(1), 96-113.

The discussion is a little too brief for a review and this could be extended by referring to some of the older literature that posed questions that may, or may not, have been answered, even in light of the significant attention that is currently be paid to human monkeypox infections. I believe that extending the introduction and discussion would do justice to this review.

The review is well written with few typographical corrections being required.

Table 2, Treatment and outcomes.  Add a close bracket in report re study [43]. I suggest placing a close bracket following pharyngotonsillitis)

Author Response

Comment 1: The author presents a succinct, informative and useful review that gathers together data from a number of papers published in recent years on human monkeypox. The review presents data in a quite readable form with clear Tables. Some studies cited contain large datasets (eg 1255 patients – Rodriguez et al 2022) with regional symptoms and severity indicated by %. These larger datasets are useful in enabling readers to reliably assess the scope of the presentations, their severity, involvement of secondary infections and the treatment outcomes. Many of the studies cited contain reports of monkeypox infections in single patient reports. While together they provide useful information on clinical cases, treatment, treatment outcomes and geographical spread thus adding to the comprehensive coverage of the condition, they are arguable less informative than the few larger studies reported. The introduction could be usefully extended by the inclusion of several other well cited papers on human monkeypox such as:

  • Brown, K., & Leggat. PA. "Human monkeypox: current state of knowledge and implications for the future." Tropical medicine and infectious disease 1.1 (2016): 8.
  • Damon, I. K. "Status of human monkeypox: clinical disease, epidemiology and research." Vaccine 29 (2011): D54-D59
  • Ježek, Z., Szczeniowski, M., Paluku, K. M., & Mutombo, M. (1987). Human monkeypox: clinical features of 282 patients. Journal of infectious diseases, 156(2), 293-298.
  • Ježek, Z., & Frank Fenner F., Human monkeypox. Monographs in Virology Vol. 17. S. Karger AG, 1988.
  • Sklenovska, N., & Van Ranst. M. "Emergence of monkeypox as the most important orthopoxvirus infection in humans." Frontiers in public health 6 (2018): 241.
  • Weaver, J. R., & Isaacs, S. N. (2008). Monkeypox virus and insights into its immunomodulatory proteins. Immunological reviews, 225(1), 96-113.

The discussion is a little too brief for a review and this could be extended by referring to some of the older literature that posed questions that may, or may not, have been answered, even in light of the significant attention that is currently be paid to human monkeypox infections. I believe that extending the introduction and discussion would do justice to this review.

Reply 1: Thank you for your insightful comments and reference recommendations. I have extended the introduction and discussion, and have incorporated all the references.

Comment 2: The review is well written with few typographical corrections being required. Table 2, Treatment and outcomes.  Add a close bracket in report re study [43]. I suggest placing a close bracket following pharyngotonsillitis)

Reply 2: Thanks for spotting the missing bracket. This has been added to Table 2.